# Life history, nest longevity, sex ratio, and nest architecture of the fungus-growing ant *Mycetosoritis hartmanni* (Formicidae: Attina)

**Ulrich G. Mueller**[1]*, **Anna G. Himler**[1,2], **Caroline E. Farrior**[1]

**1** Department of Integrative Biology, University of Texas at Austin, Austin, TX, United States of America,
**2** Department of Biology, College of Idaho, Caldwell, ID, United States of America

* umueller@austin.utexas.edu

**Data Availability Statement:** The Supporting Information contains all raw data and metadata in an Excel file, as well as the R-script used in the analyses of colony lifespan.

## Abstract

*Mycetosoritis hartmanni* is a rarely collected fungus-farming ant of North America. We describe life history and nest architecture for a *M. hartmanni* population in central Texas, USA. Colonies are monogynous with typically less than 100 workers (average 47.6 workers, maximum 148 workers). Nests occur always in sand and have a uniform architecture with 1–3 underground garden chambers arranged along a vertical tunnel, with the deepest gardens 50–70 cm deep. Foragers are active primarily between April and October. After reduced activity between November and February, egg laying by queens resumes in April, and the first worker pupae develop in early June. Reproductive females and males are reared primarily in July and August, with proportionally more females produced early in summer (protogyny). Mating flights and founding of new nests by mated females occur in late June to August, but may extend through September. For a cohort of 150 established nests (nests that had survived at least one year after nest founding), the estimated mortality rate was 0.41–0.53, the estimated average lifespan for these nests was 1.9–2.5 years, and the longest-living nests were observed to live for 6 years. These life-history parameters for *M. hartmanni* in central Texas are consistent with information from additional *M. hartmanni* nests observed throughout the range of this species from eastern Louisiana to southern Texas. Throughout its range in the USA, *M. hartmanni* can be locally very abundant in sun-exposed, sandy soil. Abundance of *M. hartmanni* seems so far relatively unaffected by invasive fire ants, and at present *M. hartmanni* does not appear to be an endangered species.

## Introduction

*Mycetosoritis* is among the least-studied genera of fungus-growing ants (tribe Attina), appearing typically as single, stray workers in surveys of ground-dwelling ants [1–14]. Since the original species description of *M. hartmanni* over 100 years ago [15], no reports have been published elucidating the biology of *M. hartmanni*, leading to the general belief that this fungus-growing ant species is exceedingly rare or difficult to find [2,16, 17].

Fungus-growing ants (subtribe Attina) are partners in an obligate symbiosis with fungi that they cultivate for food [15, 16, 18–20]. Phylogenetically, Attina ants are subdivided into two

**Funding:** The research was supported by funding from the National Science Foundation (Doctoral Dissertation Improvement Grant DEB-0206372 to AGH; CAREER award DEB-998379 and OPUS award DEB-1911443 to UGM); and the W.M. Wheeler Lost Pines Endowment from the University of Texas at Austin. The funders had no role in study design, data collection and analysis, decision to publish, or preparation of the manuscript. There was no additional external funding received for this study.

**Competing interests:** The authors have declared that no competing interests exist.

groups, a monophyletic group of higher Attina that include the well-researched leaf-cutter ants plus five genera closely related to leaf-cutters, and a second group called the lower Attina that includes less-studied, often cryptic, and phylogenetically more diverse genera. *Mycetosoritis* is one such understudied lower-Attina genus, the likely sister genus of the South American genus *Mycetarotes* [21–23].

The genus *Mycetosoritis* includes currently two described species, *M. hartmanni* [15] and *M. vinsoni* Mackay [2]. *Mycetosoritis hartmanni* ranges from the USA (Louisiana, Texas) across eastern Mexico to at least Honduras (Project LLAMA, [14]). Collections identified as *M. vinsoni* have been reported only from north-west Costa Rica [2, 6] and south-east Mexico [9], but the Mexican collections of *M vinsoni* could be misidentified *M. hartmanni*. Whether *M vinsoni* is a separate species from *M. hartmanni* is unclear [24, 25], as the morphological characters used by Mackay [2] to identify *M. vinsoni* are rather subtle and fall within the morphological variation known for *M. hartmanni* [24]. For example, workers from four nests collected by one of the authors (UGM) in 1995 in the Naranjo Valley in Parque Nacional Santa Rosa, Costa Rica, appear to be *M. hartmanni*, not *M. vinsoni* (UG Mueller, unpublished observations; [26]). Although future phylogenomic analyses may confirm the species status of *M. vinsoni*, it is also possible that the genus *Mycetosoritis* consists actually of only a single species, *M. hartmanni*, ranging from the USA to Costa Rica.

The little that is known about the biology of *M. hartmanni* derives entirely from the original species and genus description by William Morton Wheeler [15]; *hartmanni* is the type species of the genus *Mycetosoritis*). During Wheeler's last few months as faculty at the University of Texas at Austin, and before he then became curator at the American Museum of Natural History in New York, Wheeler discovered on 09. May 1903 a large population of *M. hartmanni* in sandy, rural habitat just east of the City of Austin, Texas: "There were hundreds of their nests, often within a few decimeters of one another, in the fields or in clearings among the oaks and whenever the sand was fully exposed to sun" (page 761 in [15]). Wheeler and his students spent a few days in early May 1903 excavating six nests of *M. hartmanni*. Each of the excavated nests had a single queen, about "60 to 70 workers", no alate reproductives or alate brood, and between one to three small fungus gardens. When Wheeler returned on 05. and 26. June 1903 to briefly check on this *M. hartmanni* population, the mounds of excavated soil surrounding nest entrances of *M. hartmanni* were less conspicuous, and in late June "no trace of nests could be found" (page 765 in [15]). This suggested to Wheeler that *M. hartmanni* nests are easiest to find during spring when workers increase excavation activities to expand their underground nests, and more difficult to find later when workers excavate less and mounds become inconspicuous. This seasonally restricted visibility of mounds likely contributed to the general belief that *M. hartmanni* is a rare species [2, 16, 17, 27].

In fall 1999, we found a large population of hundreds of *M. hartmanni* nests, similar to the population described by Wheeler [15], in sandy soil of semi-open pine-oak forest at the Stengl Lost Pines Biological Station, Bastrop County, Texas (Fig 1). We observed this population sporadically in early 2000, then decided in April 2000 to characterize life history and fungiculture of *M. hartmanni* throughout an entire season, with focus on estimation of sex ratios of reproductives. We also initiated in fall of 2000 a study on nest survivorship of 150 established nests (nests more than one year old) to estimate longevity and annual mortality rates of *M. hartmanni* colonies in this population.

## A note on the meaning and correct spelling of *Mycetosoritis hartmanni*

Although often written as *M. hartmani*, the taxonomically correct spelling is *M. hartmanni* (with two "nn"), because this spelling appears in the title and in the narrative of Wheeler's

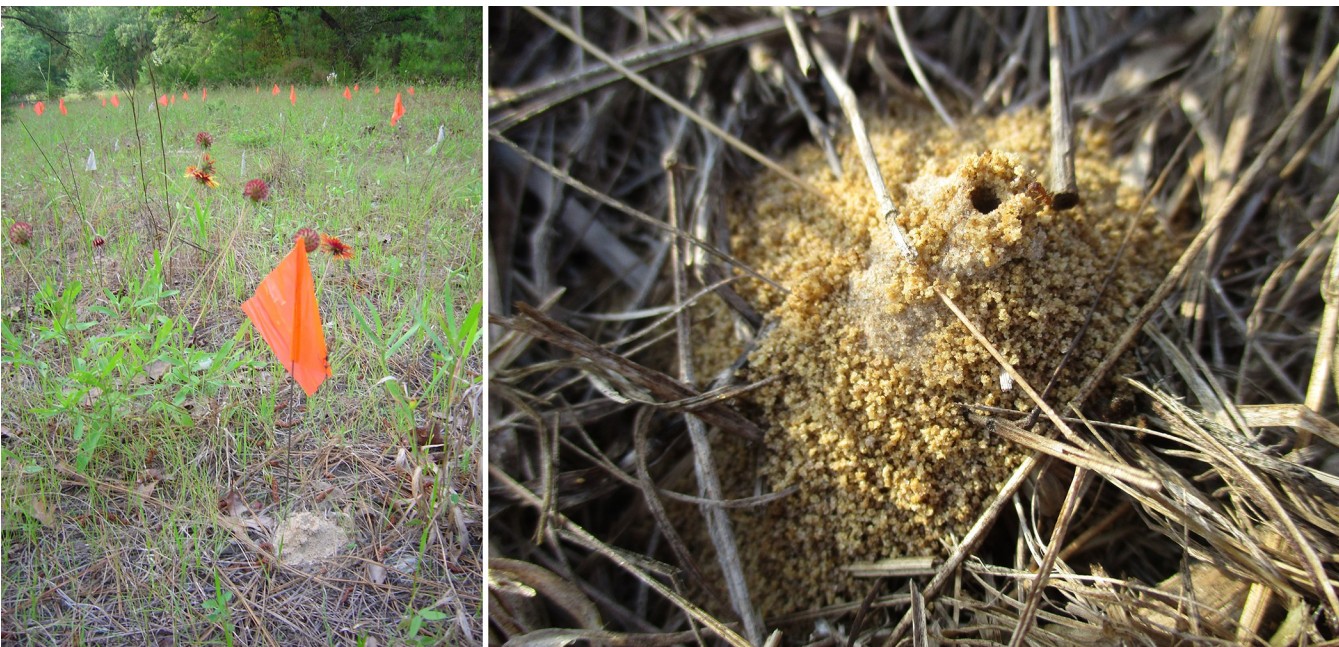

**Fig 1.** Turriform mounds of *Mycetosoritis hartmanni* at Stengl Lost Pines Biological Station (SLPBS) in Central Texas (left) and near San Manuel-Linn in the Coastal Sand Plain in South Texas (right). Both mounds shown here measure about 4 cm diameter at the base (for size comparison, the orange flag measures 8.0 cm X 6.5cm). **Left:** Our main study site at SLPBS where we excavated nests of *M. hartmanni* from April-October 2000 to assess nest architecture and chamber contents. The second site where we surveyed *M. hartmanni* nests from 2000–2007 to determine nest survivorship can be seen through the forest gap in the top-left corner of the photo. **Right:** Worker of *M. hartmanni* at the entrance of the species' characteristically steep and turriform excavation crater.

original species description of "*Atta* (*Mycetosoritis*) *hartmanni subgen. et sp. nov.*"(pages 714–717 and 761–765 in [15]). Wheeler's spelling of "*hartmanni*" was an error, as Wheeler dedicated this species to "my former pupil Mr. C. G. Hartmann" (page 716 in [15]), whose surname was actually 'Hartman', the later embryologist and reproduction biologist Carl Gottfried Hartman. Hartman had assisted Wheeler in 1903 as a Masters student, including the field work that led to the first discovery of *Mycetosoritis* nests. The misspelling "Hartmann" was a singular error in Wheeler's writing, as Wheeler later dedicated another ant species to Carl Hartman using the correct surname spelling (*Gnamtogenys hartmani*, [28]). However, given Wheeler's original writing and absence of a formal nomenclature correction, the taxonomically correct spelling is currently *M. hartmanni*, per the rules of the International Code of Zoological Nomenclature [29].

Although Wheeler [15] explained the meaning of the species name "*hartmanni*", he did not explain the etymology of the genus name *Mycetosoritis*. *Myceto-* undoubtedly refers to the fungicultural habit of this ant species. In contrast, the meaning of *-soritis* is unclear [21], but we believe *-soritis* may be latinized Greek σωρός (sorós), meaning 'heap'. Heap could refer to the steep tumulus of excavated sand typical for *M. hartmanni* mounds (Fig 1). Educated in classic languages and Greek philosophy, Wheeler and his students were likely familiar with the Sorites Paradox, the so-called 'paradox of the heap' (credited to the Greek philosopher Eubulides of Miletus, 4th century BCE), which is a paradox that arises when applying repeatedly an imperceptible change. For example, in the classical illustration of the Sorites Paradox, removing one sand grain from a heap of sand still leaves the heap, yet removing single grains repeatedly will erode the heap and leaves eventually only a single grain of sand. The paradox arises when asking then whether the single grain of sand left at the end is still a heap, and if not, when exactly did the repeated removal of single sand grains change the original heap to a non-heap? It is

possible that Wheeler intended that the genus name *Mycetosoritis* refers to 'fungus' and 'heap of sand', to allude to the fungicultural habits and the turriform sand mounds (Fig 1) characteristic of *M. hartmanni*.

## Materials and methods

All data accumulated in our study of *M. hartmanni* are in S1 Dataset.

### Study sites and nest-excavation protocol

To characterize the biology of *Mycetosoritis hartmanni* throughout a single season, we observed and excavated colonies of a single population from April to October 2000 at the Stengl Lost Pines Biological Station (Stengl LPBS), Bastrop County, Texas, USA. The Station includes pine-oak woodlands and grassy savanna, and *M. hartmanni* as well as several other fungus-growing ant species occur in white-sandy areas throughout the Station. The focal population consisted of several hundred *M. hartmanni* nests distributed across two adjacent clearings in pine-oak forest. The two dominant canopy tree species in this forest were loblolly pine (*Pinus taeda*) and post oak (*Quercus stellata*), followed by eastern juniper (*Juniperus virginiana*) and occasional blackjack oak (*Quercus marilandica*), with yaupon (*Ilex vomitoria*) dominant in the understory. The sub-population in the more southern clearing (N30.08476˚ W97.17117˚; Fig 1) was dedicated to nest excavations to assess nest architecture and chamber contents, excavating colonies regularly between April and October 2000 (sample sizes are listed in the Excel spreadsheet in S1 Dataset). The sub-population in the adjacent, more northern clearing (N30.08545˚ W97.17107˚) was reserved for a long-term study of nest survivorship between September 2000 and May 2007 (details below). Nests of *M. hartmanni* occur also in semi-open areas of the pine-oak forest surrounding the two clearings with our focal study populations, but nests were more difficult to find in forest hidden under leaf litter, while it was easy to spot the characteristic turret-like mounds of *M. hartmanni* distributed densely across the sandy ground in the open and sparsely vegetated clearings (Fig 1).

To develop a collection protocol and gain experience in excavation of *M. hartmanni* nests, we excavated three colonies on 13. April 2000. Information from these nests is excluded from the dataset analyzed here, as these excavations were imprecise, garden chambers were accidentally dug into, and many workers escaped during nest excavation. Our study therefore started with nest excavations on 20. April 2000 and continued at approximately two-week intervals until the end of August 2000, with two additional monthly surveys in September and October 2000 (see S1 Dataset). We aimed to excavate between 2–4 nests at each biweekly time point, but to improve estimation of sex ratios, we excavated seven nests in early July and six nests in late July when colonies produced most reproductives. On several occasions, additional researchers excavated *M. hartmanni* nests alongside the nests included in our study, but information from these additional nests is not included here, as these researchers lacked excavation experience and did not aim to characterize nest architecture comprehensively. For completeness, data on all nests excavated in 2000, including the practice nests that we excavated on 13. April 2000, are listed in the PhD Dissertation by co-author Anna Himler [30], but not in the datasets analyzed here (S1 Dataset), which includes only collections from researchers with sufficient experience in nest excavation of *M. hartmanni*.

In April and May 2000, we flagged about 90 nests of *M. hartmanni* in the southern subpopulation by placing surveyor flags (Glo Orange Flag Stakes, 50 cm long wire stem; Home Depot) at 5 cm distance to the north of each mound. Nests of *M. hartmanni* can be identified reliably in spring because the dark-headed workers of *M. hartmanni* can be observed on mounds throughout much of the day, colonies excavate noticeable amounts of soil in spring

after winter inactivity, and *M. hartmanni* mounds assume a characteristic appearance of steep tumuli or turrets (typically 3–8 cm diameter at the base, 2–4 cm high; Fig 1). From the sample of 90 flagged nests, we chose nests haphazardly for excavation in subsequent biweekly surveys.

During nest excavation, we measured nest architecture using a standardized protocol, which was facilitated by the stereotypical architecture of *M. hartmanni* nests. We first dug a hole (40–50 cm deep) at a safe distance of 25–35 cm to the nest entrance to avoid destruction of nest chambers or tunnels. From this hole, we then dug sideways by carefully removing the sandy soil laterally towards the tunnel subtending the entrance, starting at ground level and tracing the single narrow tunnel until we located the shallow nest vestibule (about 5 cm deep) or the first garden chamber (about 15–20 cm deep). We found that inserting a long pliable pine needle or grass stem into a tunnel minimized the chance of losing the tunnel during excavation and enabled us to trace the tunnel downward to the next chamber. We used forceps or aspirator to collect any ants found in a tunnel or chamber. We carefully extracted the garden from each chamber with forceps, and moved the garden gently into a 5-dram snap-cap vial, together with the collected workers. We traced any tunnel leading further downward from a chamber to the next chamber, and continuing downward until all chambers were excavated or the bottommost tunnel ended blindly below the deepest garden chamber. For each chamber, we recorded chamber depth (from ground surface to chamber roof), chamber dimensions (width and height, at greatest dimensions), and chamber contents (e.g., live garden, empty, excavated soil, or garden refuse temporarily stored in a chamber by workers). Because chambers were of an ellipsoid shape (width typically slightly greater than height), we later calculated chamber volumes from our measurements of chamber height and width assuming that chambers were perfectly round when viewed from above, but ellipsoid when viewed from the side. The chamber volume can then be calculated by the formula for ellipsoid volumes $(4/3)^*[\pi^*(width/2)^*(width/2)^*(height/2)]$ (see formula embedded in cells of columns G, L, Q, and V in the Excel Sheet 3 in S1 Dataset).

We censused the number of workers, queens, dealate females behaving like workers, reproductives (female or male alates) and brood for most, but not all, excavated colonies. Early in the season when nests did not contain much brood, we censused garden contents and worker numbers in the field right after collection, but starting in June, we transported collections to the laboratory to use a microscope to census brood embedded in gardens. For five of the nests that we censused in the laboratory, we also weighed male and female alates on a Mettler balance (accurate to 0.001 mg) to estimate average wet weight per male and per female alate. We used these estimates of average wet weight to convert observed numerical sex ratios into estimates of investment sex ratios (Excel Sheet 4, S1 Dataset), because wet weight is one accepted proxy of resource investment into ant reproductives [31] (see Sex Ratio below).

For comparison with the nests in the focal study population at Stengl LPBS, we include here also information on ten additional *M. hartmanni* nests that we excavated between 2001 and 2006 in Louisiana (n = 1), east Texas (n = 5) and central Texas (n = 4), using the excavation protocol described above. Exact collection locations with GPS information of these additional ten nests are listed in S1 Dataset (Excel Sheet 1). Voucher material of ants and garden from all collections are ethanol-preserved at -80°C in the Mueller Lab Collection, to be incorporated eventually in the Insect Collection of the University of Texas at Austin.

## Sex ratio

For each excavated nest with alates or reproductive brood, we calculated the numerical sex ratio (percent male) as the total number of male reproductives observed (male alates and male pupae) divided by the total number of male plus female alates and pupae (Column O in Excel

Sheet 1, S1 Dataset). Because *M. hartmanni* females are larger than males and require more resources to complete development, the numerical sex ratio underestimates actual investment of colony resources in females. We therefore weighed females (average weight of 0.7667 mg per female) and males (average weight of 0.5680 mg per male) from five different nests, as already explained above, to derive an estimated male:female weight ratio of 0.7392 (Excel Sheet 4, S1 Dataset). Multiplying this weight ratio of 0.7392 with the observed numerical sex ratios yielded estimated investment ratios for each of the *M. hartmanni* nests that had reproductives at the time of excavation (Column P in Excel Sheet 1, S1 Dataset).

## Colony survivorship

To estimate mortality and survivorship rates of colonies of *M. hartmanni*, we identified 150 nests on 30. September 2000 in the northern subpopulation at Stengl LPBS that we had reserved for a survivorship study and where we did not excavate nests. Most of these 150 nests had already been pre-flagged in spring 2000, and the time to identify and flag a total of 150 nests for our survivorship study therefore took only about 2 hours. Each of the nests included in the survivorship study was judged in fall 2000 to be more than 1 year old, based on the larger sizes of excavation craters surrounding nest entrances. We call these nests here "established nests" because these nests had survived the phase of high mortality typical for incipient or young ant nests [32]. The exact ages of these established nests was not known at the start of our multi-year survey, except that nests did not appear to be young and small (i.e., nests with small mounds were not included in our study of nest survivorship). From 2001 to 2007, we re-surveyed the cohort of 150 nests (flagged in 2000) every year in either late spring or early summer (27. May 2001; 13. April 2002; 06. June 2003; 15. May 2004; 16. May 2005; 07. July 2006; 17. May 2007) to determine survivorship until all nests had died by spring 2007 (data in Excel Sheet 6, S1 Dataset). Re-surveys in 2002–2007 were conducted blind [33] with respect to the exact number of live colonies observed during the previous year. Between years, disappearance of a flagged nest in a specific location is most likely due to colony mortality, not colony migration, because (i) we never observed any above-ground colony migration in *M. hartmanni*; (ii) it is fundamentally difficult for fungus-growing ants to relocate nests above ground because they have to move their delicate fungal gardens in addition to any brood, and transported pieces of fungus garden dry and degrade quickly above ground; and (iii) we never observed in our careful excavations of 49 nests any side tunnels that could enable any lateral below-ground migration (details below).

From these annual survivorship records between 2000–2007, we derived estimates of the annualized mortality rate ($\mu$, mortality per year) and expected lifespan of *M. hartmanni* colonies in the population at Stengl LPBS. We provide four separate estimates (Table 1) because of two different complications in these analyses. First, because we did not always census the nests on the same day of year (the initial survey of the cohort of 150 nests was in September, all later surveys of surviving nests were between April and July in subsequent years; see above), and because mortality events may differ between seasons (e.g., mortality may be greater in winter because of extreme cold spells; or greater in summer because of heat stress and desiccation; or greater in spring when army-ant predators may be most active), we generated estimates making alternative assumptions covering the extreme possibilities: (i) mortality was assumed to occur only outside the re-survey season April-July (Estimates 1 and 3); or alternatively (ii) mortality risk was assumed to be constant across all days of the year including the re-survey months April-July (Estimates 2 and 4) (Table 1). The second complication in our analyses is a sample-size complication inherent in survivorship data. Across each census interval, we observed survival and mortality for samples of individual nests, and sample sizes were

**Table 1. Mortality rate, expected lifespan, and maximum lifespan estimated from observed survivorship records of a cohort of 150 *M. hartmanni* colonies from a single population at Stengl Lost Pines Biological Station surveyed annually in late spring or early summer between 2000–2007 until all colonies had died.** Maximum lifespan is defined as the timepoint at which 95% of colonies of a cohort are expected to have died. The four Estimates 1–4 are based on models that differ in modeling assumptions. Mortality was assumed to occur only outside the annual re-survey season April-July (Estimates 1 and 3); or alternatively, mortality risk was assumed to be constant across all days of the year including the re-survey months April-July (Estimates 2 and 4). Estimates 1 and 2 ignore differences in sample sizes across years (later years have naturally smaller sample sizes because of nest mortality), while Estimates 3 and 4 weigh confidence in the mortality rates across years by the number of individual nests at the start of a given census interval (additional details in Materials and methods).

| Estimate | Model assumptions | Mortality rate ($\mu$, colony death rate per year) | Expected lifespan of established colonies (years) | Maximum lifespan of established colonies (years) |
|---|---|---|---|---|
| 1 | mortality only in summer-to-winter, no weighting by sample size | 0.51 | 1.95 | 5.84 |
| 2 | constant mortality each day throughout year, no weighting by sample size | 0.53 | 1.89 | 5.67 |
| 3 | mortality only in summer-to-winter, weighting by sample size | 0.41 | 2.46 | 7.37 |
| 4 | constant mortality each day throughout year, weighting by sample size | 0.47 | 2.13 | 6.39 |

naturally much larger at the beginning of the study (initially 150 nests) than later in the 7-year survey when the number of surviving nests gradually converged to zero. We found in our modeling that the sample-size differences generated small discrepancies in our estimates of mortality. Estimates 1 and 2 ignore the differences in sample sizes across year, while Estimates 3 and 4 weigh our confidence in the mortality rates across years by the number of individuals at the start of a given census interval. An even better statistical method would be to fit the exponential decay in counts (i.e., surviving colonies), but we believe that this would be less accurate because of the necessity of assuming exact time points of mortality (e.g., winter or summer) to fit such models. We therefore present here only the four estimates derived from the above four different analyses incorporating different mortality-risk assumptions and different sample-size weighing schemes, as explained above.

From our estimates of mortality ($\mu$, mortality per year; Table 1), and assuming that mortality is constant with age, we calculate the expected lifespan ($1/\mu$, the inverse of $\mu$; i.e., the mathematical expectation of an exponential distribution) and expected "maximum" lifespan as the age of the nests when less than 5% of an original cohort is expected to have survived [$-\mu^{-1} \ln(0.05)$, in years]. Constant annual colony survival likely approximates true survival rate for much of the life of an established colony, but colony survivorship likely declines and mortality risk of a colony may increase very late in a colony's life whenever a queen may senesce, or a queen may run out of stored sperm in rare cases and can no longer produce workers, leading to inevitable colony death as workers die out. Queen senescence or sperm shortage in *M. hartmanni* are unlikely to affect colony survivorship already at our estimated average (expected) lifespan for established colonies of approximately 1.9–2.5 years ($1/\mu$; Table 1), because ant queens are selected for efficient sperm storage and for extraordinary long life compared to workers [34, 35], but sperm depletion may be possible at ages close to the maximum lifespan. Our estimate of expected (average) lifespan for established colonies of *M. hartmanni* is therefore likely relatively unbiased, while our estimate of maximum lifespan may somewhat overestimate actual maximum lifespan of established *M. hartmanni* colonies.

## Results and discussion

### Sample sizes and collection biases

Between April and October 2000, we excavated and censused at Stengl Lost Pines Biological Station (Stengl LPBS) a total of 39 established *M. hartmanni* nests (estimated age of more than

1-year old), excavating between 2–6 nests at typically 2-week intervals during this time period (exact sample sizes and excavation dates are listed in Excel Sheet 1 in S1 Dataset), with more nests excavated during June and July when nests produced reproductives. We also excavated at Stengl LPBS two young nests that were approximately one year old (excavated June and July 2000), and two incipient nests founded by females that swarmed in 2000 and that were approximately 1–3 months old at the time of nest excavation in August and September 2000. In addition, we excavated ten established nests of *M. hartmanni* between 2001–2006 in Louisiana (n = 1), east Texas (n = 5), and central Texas (n = 4) (Excel Sheet 1, S1 Dataset) during general collecting of ants. The preponderance of larger, established nests in our collections was intentional because, for estimation of sex ratios, we tried to avoid excavation of younger nests (as judged by mound sizes), because smaller and younger nests are less likely to produce reproductives.

## Seasonal worker activities, habitat, and general behavior

Throughout the range of *M. hartmanni*, mounds were easiest to find in spring when workers increase excavation activities and all nests have conspicuous mounds of excavated sand, as already noted by Wheeler [15] (see above). Mounds are frequently steep accumulations of sand (3–8 cm diameter, 2–4 cm high, rarely somewhat larger) (Fig 1). Such "turriform crater of pure sand" (page 761 in [15]) are diagnostic for *M. hartmanni* (i.e., no other ant builds such turriform crater in Texas; [36]). Nests can be more difficult to find in summer through winter, and after rains when nest entrances become flushed shut with sand and workers sometimes need days to re-open nest entrances. Small entrance holes of *M. hartmanni* (≈1.5 mm diameter) with little or no excavate can be found in summer through winter, and these nest entrances are then difficult to distinguish from those of other ant species. For example, when surveying our study populations at Stengl LPBS during the early afternoon on 23. March 2002, only few of the flagged *M. hartmanni* nests had fresh excavate at entrance mounds, entrances of most nests were closed, no workers were seen at any of the nests despite warm afternoon temperatures of about 20°C, and the presence of a large population of *M. hartmanni* nests at this site would have remained unknown to a naïve observer. The steep entrance tumuli identifying *M. hartmanni* nests (Fig 1) are easiest to find after a time of no rain during the second half of spring when colonies excavate most sand during nest expansion. This explains why we found during general ant collecting throughout Texas additional *M. hartmanni* populations frequently in spring, and less during other seasons. Throughout the range of *M. hartmanni* in the USA, nests occur essentially only in white sand (rarely in red sand), typically in sun-exposed areas with sparse or no ground vegetation, like the habitat of the two forest clearings described above for the main study populations at Stengl LPBS.

Outside the nest, workers of *M. hartmanni* appear timid; when disturbed, workers feign death readily by curling up into a motionless ball, a typical defense response of many lower-Attina species. Workers feigning death are then difficult to distinguish from particles on the ground. Wheeler (page 764 in [15]) believed that "only few workers [of a *M. hartmanni* nest] forage or excavate at a time" and that "the ants appear to be nocturnal or crepuscular". We agree that it is less likely to observe excavating or foraging workers at nests exposed to full sunshine, but we observed that other *M. hartmanni* nests nearby that are shaded may at the same time show above-ground worker activities. On hot days, therefore, workers appear to avoid exposure to full sun, but workers can be active in shaded areas throughout the entire day, so it would seem incorrect to call *M. hartmanni* a nocturnal or crepuscular species.

Near the southern range limit of lower-Attina ants in southern Brazil, at a latitude of 29° South that is comparable to the latitude of 30° North of the *M. hartmanni* population at Stengl

LPBS, nests of the lower-Attina ant *Mycetophylax clorindae* (formerly *Mycetosoritis clorindae*; [21]) can enter a state of hibernation during the austral winter. In a hibernating nest of *Mycetophylax clorindae*, only a small cohort of 10–15 workers is active, and these workers plant fungal mycelium on the queen and inactive workers, such that most of the workers and the queen are embedded in curled-up, motionless postures in the protective garden matrix [37]. To test whether *M. hartmanni* nests enter a similar state of hibernation during the boreal winter near the northern range limit of lower-Attina ants, we excavated two nest of *M. hartmanni* on 27. November 2004 (late fall) and 23. February 2005 (winter). None of the workers and none of the queens of these two *M. hartmanni* nests were embedded and immobilized in the garden matrix, and workers and queen moved freely about the gardens as they do in spring and summer. It is possible that *M. hartmanni* nests enter hibernation during winters that are colder than the unusually warm winter of 2005. A lab experiment exposing *M. hartmanni* nests to gradually colder temperatures could test for possible winter adaptations like the peculiar fungus-embedding of queen and workers observed in *Mycetophylax clorindae* during the austral winter [37].

## Colony size

Established colonies at Stengl LPBS had an average of 47.6 (± 31.9 SD) workers per nest, most colonies had counts between 20–70 workers, only two colonies had counts over 100 workers (118 and 148 workers) (Excel Sheet 2, S1 Dataset). These counts underestimate actual colony sizes somewhat, we estimate by about 5–15%, because some workers likely escaped during nest excavation (particularly early in the 2000 season when we had less experience excavating nests) and because some foraging workers may have been absent from a nest at the time of excavation. The two young colonies of about 1-year age had counts of one and four workers, respectively; and the two incipient colonies had no workers when we excavated these approximately 1–3 months after mating flights. The observed colony size of 47.6 workers per *M. hartmanni* nest at Stengl LPBS is somewhat less than the estimated 60–70 workers per nest reported by Wheeler [15] from a nearby population in central Texas, and less than the colony size of 66.2 workers (± 43.2 SD) observed in ten additional *M. hartmanni* nests excavated in western Louisiana and east & central Texas (Excel Sheet 2, S1 Dataset). The largest nest of *M. hartmanni* had a count of 148 workers, but counts per nest were typically less than 100 workers (Excel Sheet 2, S1 Dataset). Many other fungus-farming ant species of diverse lower-Attina genera also have colony sizes of less than 100 workers [16, 18, 21, 37–45].

## Number of queens, dealate females behaving like workers, and queen loss

Queenright colonies of *M. hartmanni* had always one queen, which could be identified as the single female with the darkest integument and a characteristic behavior of hiding in garden when a nest was disturbed. The tendency of queens to hide in a garden, rather than escaping when a nest is disturbed, explains our high success rate of finding a queen in almost all nests. Five nests also had dealate females that behaved like workers and that tended to have a lighter integument than queens, indicating a younger age. Dealate females exhibiting worker behavior like foraging, nest excavation, or rescuing brood are known from many other fungus-farming ant species [16, 18, 40, 46–48]. Dealate females in nests of fungus-farming ants are thought to be females that fail to disperse from the natal nest, shed their wings, and then assume worker duties in their natal nest. We observed such transitions from alate to dealate females in our laboratory colonies of *M. hartmanni* where alates are unable to swarm and mate. One nest (UGM000721-01; Excel Sheet 1, S1 Dataset) appeared to be queenless (we found no queen in this nest), and this nest produced an unusually large number of 163 males, but no female

reproductives and no worker brood. This suggests that either (a) the queen ran out of sperm to fertilize eggs and we failed to collect this queen during our nest excavation; or alternatively (b) unmated workers may have laid unfertilized eggs after queen loss, leading to the extreme excess of males.

## Semi-claustral nest founding

One foundress queen was observed on 31. August 2000 at 8:15 AM returning from foraging to her incipient nest, a small mound of about 20 mm diameter and 8 mm high, and we were then able to excavate this nest (UGM000831-01) to verify that this was an incipient nest. The nest must have been founded by the queen during the early mating flights in July or possibly August 2000 (see below), and this incipient nest therefore was less than two months old. At the time of excavation at the end of August, we found no workers in this nest and only the single queen, curled up in a defensive posture at the bottom of a small foundress chamber (1.0 cm wide, 1.2 cm high) that the queen had excavated at a depth of 14 cm from ground level (Excel Sheet 1, S1 Dataset). The queen had produced a small garden (≈8 mm diameter) hanging from the ceiling of this foundress chamber. We found no large larvae or pupae in the incipient garden, but small brood may have been present hidden in the garden matrix. The observation of foraging behavior by a foundress queen suggests semi-claustral nest founding, as occurs in other fungus-farming ant genera but not in the genus *Atta* [16, 18, 49–51].

## Colony survivorship

The linear relationship between time and log(colony survivorship) observed in the cohort of 150 established nests that we tracked annually between 2000–2007 (Fig 2) is consistent with a

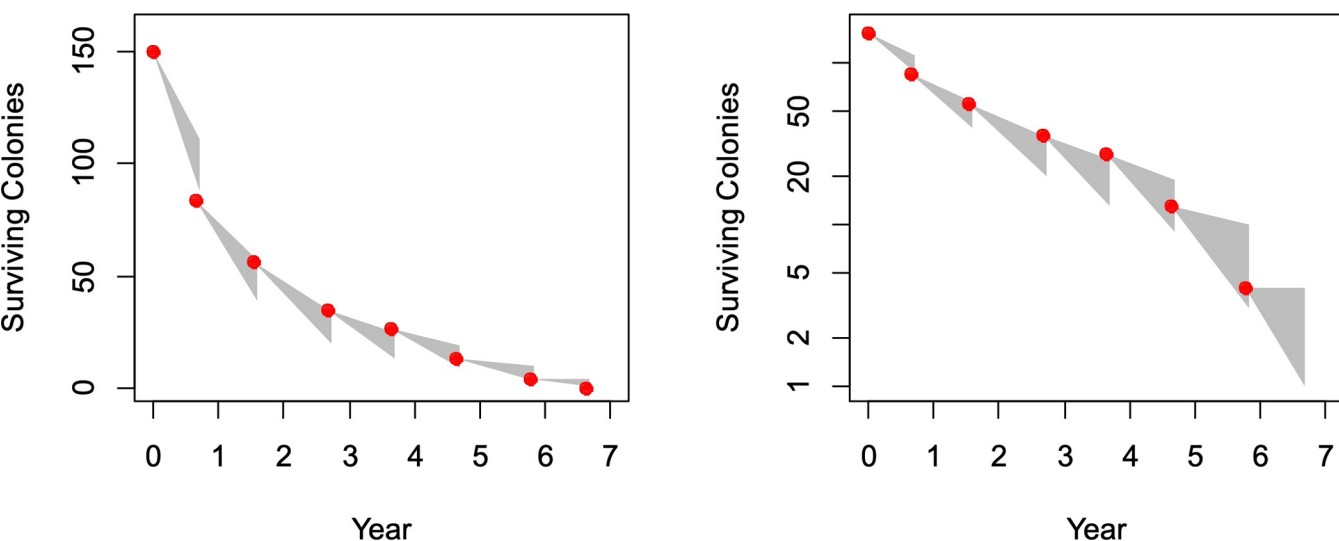

**Fig 2.** Colony survivorship in a cohort of 150 established colonies of *Mycetosoritis hartmanni* observed between 2000–2007 (Year 0 to Year 7), plotted as the absolute number of surviving colonies over time (left) and as the Log of the number of surviving colonies (right). Each gray triangle shows the 95% confidence interval of the estimate of the number of surviving colonies projected for the next census (height of the right edge of each triangle), given the number of surviving colonies observed in the previous census (left apex of each triangle). The graphs shown here are for Estimate 4 in the modeling analyses of colony mortality; graphs for all four Estimates 1–4 are in S1 File. All four modeling estimates show the exponential decline of the number of surviving colonies (left) and the approximate linear decline in the Log of the number of surviving colonies (right), suggesting a constant mortality rate over much of the life of an established colony, except perhaps in the very oldest colonies. The exact age of each colony in 2000 was not known, but each of the 150 colonies seemed established at that time (not a small excavation crater at the nest entrance) and therefore each colony was at least one year old in 2000. The estimated mortality rate was 0.41–0.53 for the cohort of 150 surveyed colonies, and the estimated expected (average) lifespan was 1.9–2.5 years for these colonies (Table 1). The longest-living colonies lived for 6 years after the start of our survey, consistent with the estimated maximum lifespan of 5.7–7.4 years for established colonies (Table 1).

constant mortality rate once *M. hartmanni* colonies had survived the fragile nest establishment phase, accumulated sufficient workers to safeguard and maintain a colony, and transitioned to an established colony (defined here operationally as a colony that was older than one year). This suggests a Type II survivorship for the duration between nest establishment early in colony life and nest senescence late in colony life when a queen may senesce physiologically and may no longer produce workers to maintain a colony. Between 2000–2007, the estimated mortality rate was 0.41–0.53 for the cohort of 150 surveyed nests, and the estimated expected (average) lifespan for these nests was 1.9–2.5 years (Table 1). The longest-living nests in our survey lived for 6 years after the start of our survey, consistent with the estimated maximum lifespan of 5.7–7.4 years for established colonies (Table 1). Lifespan of *M. hartmanni*, estimated here for the first time for any non-leafcutter fungus-growing ant species, is therefore markedly shorter than the lifespan of leafcutter ant colonies, which are thought to live for 10–20 years and perhaps longer in some rare cases [52–56].

## Absence or rarity of colony migration

The preceding colony-survivorship analyses presuppose that disappearance of a flagged nest in a specific location is due to colony mortality, not colony migration. This presupposition is justified for three reasons. First, we never observed any above-ground colony migration in *M. hartmanni* during our observations of nest aggregations throughout the US range of this species. Second, it is fundamentally difficult for fungus-growing ants to relocate nests above ground because they have to move their delicate fungal gardens in addition to any brood, and because pieces of fungus garden carried by workers dry and degrade quickly above ground. These reasons suggest absence or rarity of above-ground colony migrations in *M. hartmanni*. Third, we never observed in our careful nest excavations any side tunnels that could enable any lateral below-ground migration (i.e., each of the 49 established nests that we excavated had only a single vertical tunnel linking garden chambers, never any side tunnels), suggesting absence or rarity also of below-ground colony migrations. Disappearance of nests in our long-term survey of 150 established nests is therefore very unlikely due to colony migration, but due to colony death.

## Brood production, time of mating flights, and incipient nests

Because we did not find larvae in gardens of *M. hartmanni* collected in fall, winter or early spring, and because Wheeler [15] likewise did not find any brood in early May, queens appear to cease reproduction during the colder seasons in central Texas, and egg-laying by queens starts annually in late April or early May and continues until at least August (Excel Sheet 1, S1 Dataset). The first worker pupae appeared in our collections in early June and the first reproductive pupae in late June, suggesting that the first brood produced annually by colonies develop into workers. We found the first eclosed alate reproductives (alates) in nests in early July, the great majority of alates in July and August, very few alates in late September, and no alates in later nest excavations. The primary months for mating flights of *M. hartmanni* in central Texas are therefore July through September, and most new nests are presumably founded during those months following regular mating flights. We never observed a mating flight of *M. hartmanni* during the mornings and early afternoons when we were present at our main study site to excavate nests, so mating flights of *M. hartmanni* may occur at dawn, during the evening, or possibly at night.

Because July and August are the hottest and driest months in central Texas [57], with more regular rains resuming typically in late August or early September, and because garden chambers of newly founded nests are shallow (about 15 cm deep; Excel Sheet 1, S1 Dataset), nest

mortality of newly founded nests is likely high, particularly in years with dry summers and falls. Interestingly, of two incipient nests that we excavated in late August and late September (Excel Sheet 1, S1 Dataset), only the nest excavated in August had a healthy garden. The other incipient nest excavated in late September had no garden, possibly because the foundress queen of this incipient nest had either very recently dug her nest and had not yet started her garden, or alternatively this foundress queen lost her garden because of drought, garden disease, or unknown factors.

The average count of workers in the 15 nests where we found alate reproductives was 41.8 (± 20.6 SD), and of these 15 nests, the four nests with fewest worker counts had between 17–24 workers (Excel Sheet 5, S1 Dataset). Approximately 20 workers may therefore be the colony size minimally required for alate production, but this estimate of 20 workers is likely an underestimate of the true colony-size threshold for alate production because some workers escaped during our nest excavations and some foraging workers may have been absent from nests at the time of our excavations (see above).

The number of workers found in each of the 15 nests with alate reproductives was significantly positively correlated with the respective number of total alate brood found in each nest (Pearson correlation coefficient $r = 0.562$, $p = 0.0291$), but not statistically significantly correlated with the respective total number of eclosed alate reproductives found in each nest (Pearson correlation coefficient $r = 0.4152$, $p = 0.1238$) (Excel Sheet 5, S1 Dataset). Because alate reproductives depart regularly from nests during mating flights, it is perhaps no surprise that worker number was not correlated with the number of alate adults found in each nest, even though we found evidence that larger nests with more workers produce more alate brood ($r = 0.562$, $p = 0.0291$).

## Sex ratio

We found pupae of reproductives only between late June and late July, and alate reproductives in nests only between early July and late September, with the great majority of reproductive found in nests in July and August (Fig 3). The estimated wet weight of a female (0.767 mg) is about 38% more than the wet weight of a male (0.568 mg), that is, the average male weighs only about 74% the weight of an average female (Excel Sheet 4, S1 Dataset). Consequently, numerical sex ratios overestimate male investment by about 15% (Excel Sheet 1, S1 Dataset) (Fig 3). Combining counts from both reproductive pupae and alates observed in single nests (columns L-N in Excel Sheet 1, S1 Dataset), the numerical and investment sex ratios shifted from more female production early in the season (late June & early July; sex ratios of 20–30% males) to predominantly male production later in the season (late July to September; sex ratios of 60–90% males) (Fig 3). *M. hartmanni* therefore exhibits a protogynous sex-ratio transition from female-biased to male-biased sex ratios during the reproductive season, similar to the shift from early predominance of female atales to later predominance of male alates observed in excavated nests of *Atta cephalotes* and *A. sexdens* in Suriname [58]. Moreover, because the first brood that is produced in nests in spring (late May and early June) consists entirely of worker brood, and because worker production in *M. hartmanni* colonies is reduced or may even cease in some colonies during the phase when reproductives are produced (Excel Sheet 1, S1 Dataset), the protogynous sex-ratio indicates also a transition by colonies that initially in spring rear diploid brood only into workers, and somewhat later rear diploid brood preferentially into female reproductives. This annual transition in spring from investment into colony growth and maintenance (investment into workers) versus into reproduction could be under queen control, or alternatively could be under worker control because workers influence rearing conditions (e.g., nutrition, temperature) of developing female larvae [31].

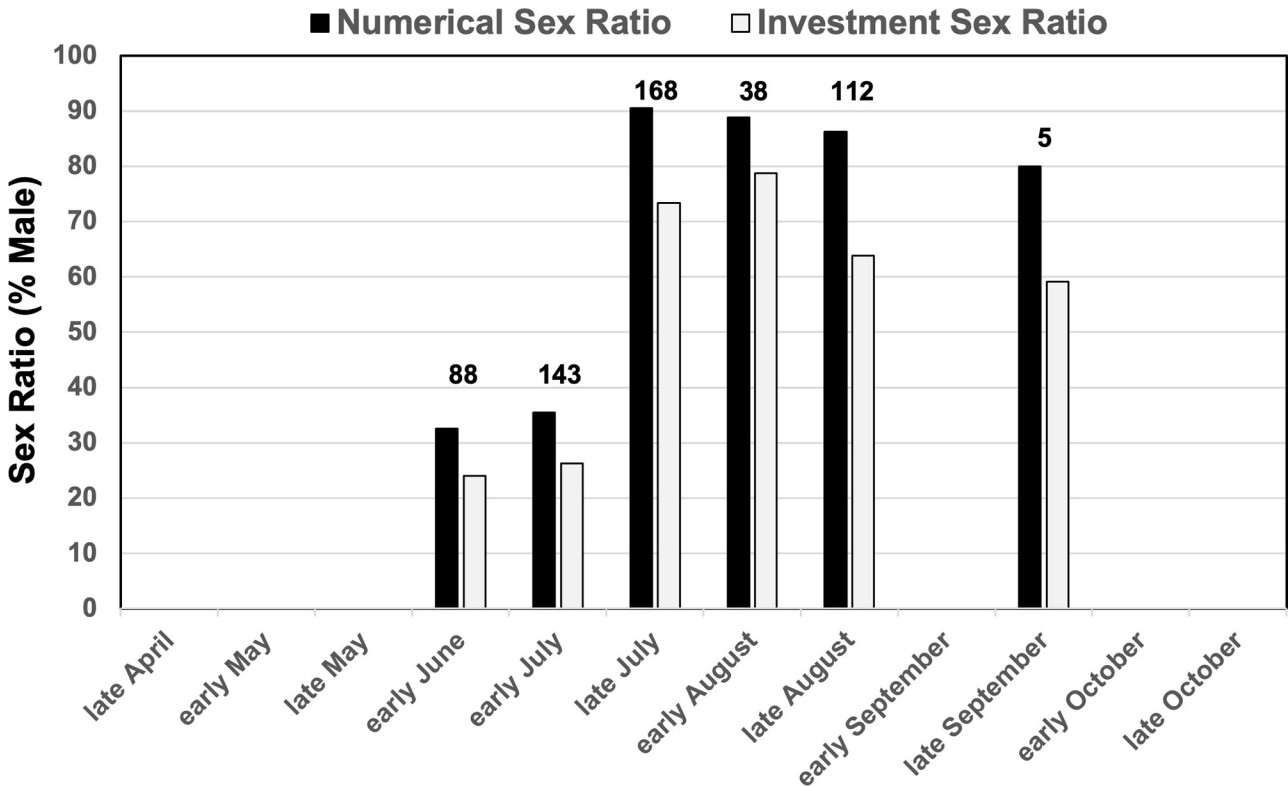

**Fig 3. Sex-ratio changes of *Mycetosoritis hartmanni* colonies collected throughout one reproductive season in 2000 at Stengl Lost Pines Biological Station.** Numerical sex ratios (black) and investment sex ratios (light grey) are shown for nests collected during the first two weeks (early) or last two weeks (late) of each month when reproductive (females + males) were found in nests. Counts of reproductives present in a nest include both reproductive pupae as well as adult alates. No reproductives were found in nests collected in April, May, and late October; no nests were collected in early September and early October. Numbers above bars indicate the total number of reproductives that were found in nests excavated in a given two-week time period. Average sex ratios were calculated by first calculating the sex ratio for each individual nest, then averaging sex ratios across all the nests collected within a 2-week time period (Excel Sheet 1 in S1 Dataset). *M. hartmanni* nests had alate reproductives only between early June and late September, and most alates were produced in July and August. Proportionally more females were produced early (protogyny) and proportionally more males later in the season. *M. hartmanni* therefore exhibits a protogynous transition from female-biased to male-biased sex ratios during the reproductive season.

The number of workers found in each of the 15 nests with alate reproductive was not statistically significantly correlated with respective numerical sex ratios (Pearson correlation coefficient $r = 0.049$, $p = 0.862$) or with investment sex ratios (Pearson correlation coefficient $r = 0.096$, $p = 0.733$) (Excel Sheet 5, S1 Dataset). Given the protogynous sex-ratio transition of *M. hartmanni* during the season (Fig 3), and given that both large and small nests undergo this transition, the finding of no overall correlation between worker number and sex ratio is perhaps no surprise, because nests with few and with many workers were excavated early in the season (female-biased sex ratios) and also late in the season (male-biased sex ratios), obscuring any subtle effects that colony size may have on sex ratio.

Between the 15 nests of *M. hartmanni* in which we found either adult sexual alates or sexual pupae at the time of excavation (Excel Sheets 1 & 5, S1 Dataset), we collected a total of 180 females and 537 males, suggesting a numerical sex ratio of 74.9% calculated across the entire season and a corresponding investment sex ratio of 55.4% (using the weight conversion factor explained above). Of the 537 males, 163 males derived from the aforementioned queenless nest UGM000721-01 in which the queen appeared to have died and workers may have produced all males. Excluding this queenless nest from the sample used for calculation of

population sex ratio across the entire season, we found 180 females and 374 males between the remaining 14 nests with sexual brood or alates at the time of excavation, suggesting a numerical sex ratio of 67.5% and a corresponding investment sex ratio of 49.9% (Excel Sheets 5, S1 Dataset). Both analyses (with or without the queenless nest) suggest an investment sex ratio across the entire season of about 50%, consistent with standard Fisherian sex-ratio evolution [31]. We found therefore no evidence that sex ratios of *M. hartmanni* are biased towards females by either the workers or by the cultivated fungus. Workers are predicted to bias sex ratios towards their more closely related sisters and away from the more distantly related brothers, resulting in queen-worker conflict over the sex ratio in monogynous nests with single-mated queens [31]. The cultivated fungus may bias sex ratios towards females, because only females propagate the fungus to the next generation of nests (i.e., males are a waste of investment from the perspective of the cultivated fungi), resulting in ant-fungus conflict over the sex ratio [59]. The population investment sex ratios of about 50% observed here for *M. hartmanni* do not support either prediction of worker- or fungus-induced female bias. For 30 Attina species for which sex ratios have been estimated to date (this study and [59–62]), only a few estimates suggest possible female-biased sex ratios. Overall across the Attina, however, there is insufficient evidence for sex-ratios deviating from the equal investment into the two sexes predicted by standard Fisherian sex-ratio evolution [31], suggesting general absence of worker control and absence of fungus control over sex ratios of Attina ant hosts.

## Nest architecture

*M. hartmanni* has a uniform nest architecture, almost stereotypical, which facilitated our nest excavations because we were able to anticipate during excavations the depths of fungus chambers. As already noted by Wheeler [15], *M. hartmanni* nests have a diagnostic turriform crater (Fig 1). From the single narrow nest entrance at the apex of this turriform crater, each colony digs a single, non-branching tunnel of about 1.5–2.0 mm diameter vertically straight down, and between 1–3 garden chambers that are strung along this descending tunnel like "beads on a thread", with tunnels that "are uniform, tubular passages entering and leaving the chambers at rather definite points" (page 199 in [63]). We never found a nest with more than three garden chambers. The descending tunnel always enters a chamber at the chamber roof, and always continues downward from the floor of a chamber; that is, chambers are never linked to the main descending tunnel via horizontal side tunnels, as occurs in some other lower-Attina ant species [16, 23, 43, 48, 64]. The entrance to a tunnel descending from a chamber floor is often surrounded by a button-like shallow cone of soil that is built up from the chamber floor around the entrance to the descending tunnel. The function of such a button-like cone is unclear.

At Stengl LPBS, the two foundress chambers that we observed in two incipient nests at an average of 15.0 cm depth (1.0 cm width, 1.1 cm height) were at about the same depth as the first garden chamber observed in established nests (average of 16.3 cm depth; 1.5–4.0 cm wide, 1.5–3.5 cm high; Fig 4, Table 2; Excel Sheet 3, S1 Dataset), suggesting that colonies expand the foundress chamber into the first garden chamber of older nests. We found second chambers at typically 25–35 cm depths (1.5–5.5 cm wide, 1.0–4.0 cm high) and third chambers at typically 40–60 cm depths (1.5–6.5 cm wide, 1.0–6.0 cm high) (Fig 4). The observed chamber depths at Stengl LPBS are similar to the chamber depths observed in the ten additional nests excavated in 2001–2006 in Louisiana and east & central Texas, but are somewhat more shallow than the chamber depths reported by Wheeler [15] for a population of *M. hartmanni* nearby in central Texas (Fig 4, Table 2; Excel Sheet 3, S1 Dataset). The deeper chamber averages reported by Wheeler [15] (Fig 4, Table 2) may be because of the specific soil conditions at the location

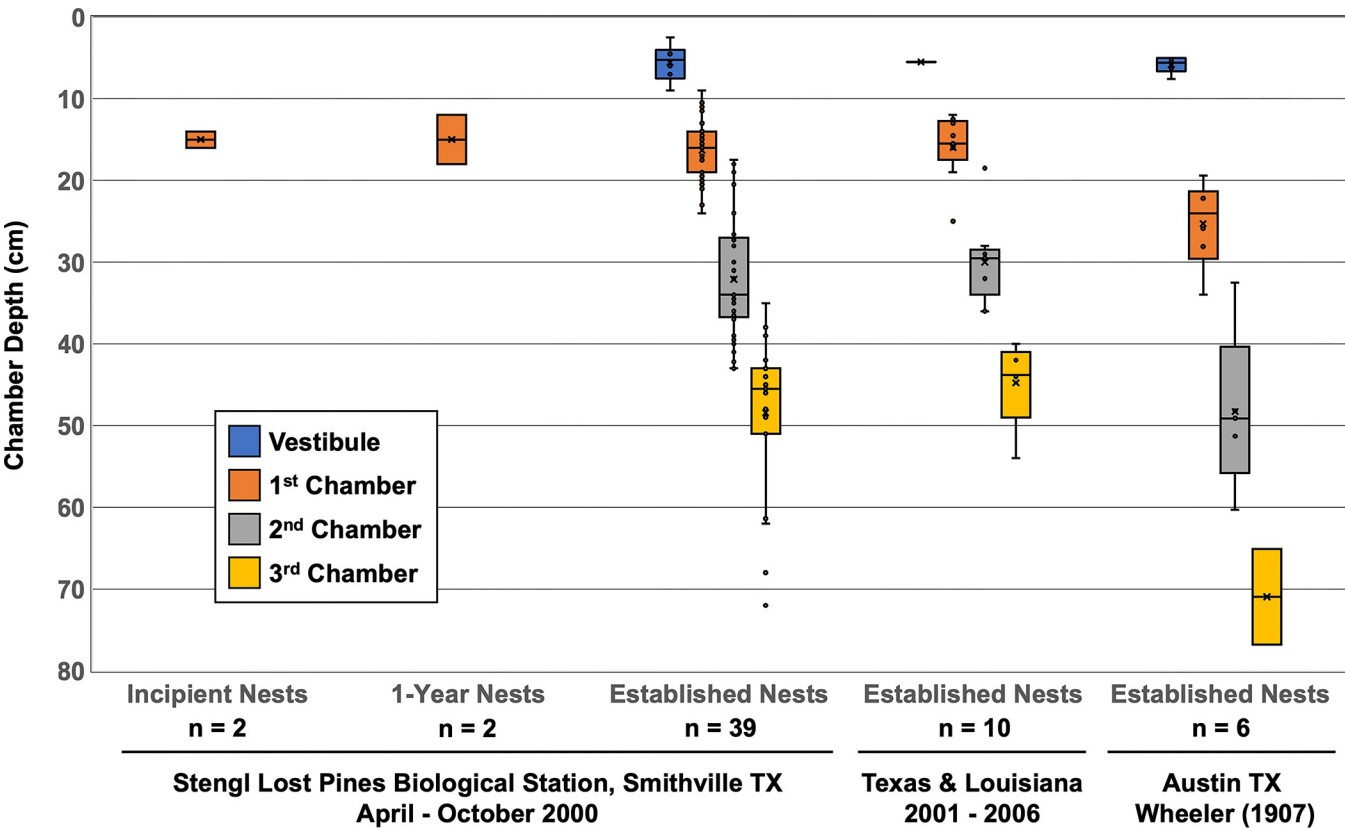

**Fig 4. Nest architecture of *Mycetosoritis hartmanni*, for incipient nests (n = 2), 1-year-old nests (n = 2), and established nests (n = 39) excavated at the Stengl Lost Pines Biological Station in Smithville, Texas; at other locations in central & east Texas and in western Louisiana (n = 10); and in Austin, Texas (Montopolis; n = 6) by Wheeler [15].** Chamber depths are measured in centimeter (cm) from the ground surface to chamber roofs, and color-coded for the Vestibule (blue, ≈5 cm deep; only some established nests have a vestibule; young nests do not appear to construct a vestibule), the 1st Chamber (red, 15–20 cm deep), the 2nd Chamber (gray, 25–35 cm deep), and the 3rd Chamber (yellow, 40–60 cm deep). Average chamber dimensions and volumes are summarized in Table 2; all raw data are in the Excel file in S1 Dataset. Incipient nests have a foundress chamber at 12–18 cm depth, and this foundress chamber is expanded into the 1st garden chamber (red) in older nests. We did not find any nest with more than 3 garden chambers, but such nests may occur at very low frequencies. The deepest chambers found were 65–75 cm deep. Nests excavated at Stengl Lost Pines Biological Station (LPBS) tended to have more shallow garden chambers compared to the nests excavated by Wheeler [15] in nearby Austin (Montopolis) ≈50 km west of the Stengl LPBS.

where Wheeler excavated colonies of *M. hartmanni* in 1903 (Montopolis, ≈50 km west of our study site at Stengl LPBS), climate change during the last century affecting nest excavation behavior of *M. hartmanni*, or measurement error (Wheeler is known to have made small calculation errors when converting measurements recorded in inches to metric measurements for publications; see one such calculation error by Wheeler discussed below under Fungiculture).

Underground chambers of *M. hartmanni* tend to be ellipsoid, somewhat wider than high, but there are exceptions (Excel Sheet 3, S1 Dataset). Chamber volumes increase with depth from typically grape-sized first chambers (volume ≈10–20 cm$^3$) to typically walnut-sized second and third chambers (volume ≈20–40 cm$^3$) (Table 2; Excel Sheet 3, S1 Dataset), but a few exceptionally large second and third chambers had volumes of 60–90 cm$^3$. The size of the third chamber likely depended on colony age or size, with younger or smaller colonies having smaller third chambers because the workers added a third chamber only recently to a nest and this third chamber had yet to be enlarged by the workers. The largest garden chamber found, an unusually large third chamber, had an estimated volume of 93 cm$^3$, about the size of a large chicken egg (Excel Sheet 3, S1 Dataset).

**Table 2. Nest architecture of *Mycetosoritis hartmanni*.** Statistics summarized are averages (± standard deviations) of chamber dimensions (width, height), estimated chamber volumes, and chamber depths from ground level of nests excavated between April—October 2000 at the Stengl Lost Pines Biological Station; in 2001–2006 in western Louisiana (LA) and eastern & central Texas (TX); and in 1903 in Austin Texas by Wheeler [15]. Garden chambers tend to be of ellipsoid shape, with width slightly exceeding height of a chamber. Incipient nests and 1-year-old nests had only one chamber, and the absence of any additional chambers in these nests is indicated by "–". In some cases, only a single measurement was available for a nest category, so standard deviations could not be calculated, indicated by (± n/a). The complete dataset from which we calculated summary statistics of nest architecture is in Excel Sheet 3 in S1 Dataset.

| | Incipient Nests | 1-Year Nests | Established Nests | Established Nests | Established Nests |
|---|---|---|---|---|---|
| | Stengl Lost Pines Biological Station, Apr–Oct 2000 | | | TX & LA 2001–2006 | Wheeler (1907) |
| **Sample Size** | n = 2 | n = 2 | n = 39 | n = 10 | n = 6 |
| **Vestibule** | | | | | |
| Width in cm | – | – | 1.5 (± 0.44) | 0.8 (± n/a) | 1.7 (± 0.48) |
| Height in cm | – | – | 1.5 (± 0.49) | 1.1 (± n/a) | 1.3 (± 0.39) |
| Volume in cm$^3$ | – | – | 2.0 (± 1.92) | 0.4 (± n/a) | 2.3 (± 2.17) |
| Depth below ground in cm | – | – | 5.6 (± 2.27) | 5.5 (± n/a) | 5.9 (± 1.02) |
| **First Chamber** | | | | | |
| Width in cm | 1.0 (± 0.00) | 1.1 (± n/a) | 2.3 (± 1.05) | 2.1 (± 0.93) | 3.8 (± 0.31) |
| Height in cm | 1.1 (± 0.14) | 0.8 (± n/a) | 2.4 (± 0.82) | 1.9 (± 0.81) | 2.4 (± 0.75) |
| Volume in cm$^3$ | 0.6 (± 0.07) | 0.5 (± n/a) | 10.1 (± 10.41) | 6.7 (± 8.69) | 18.4 (± 6.94) |
| Depth below ground in cm | 15.0 (± 1.41) | 15.0 (± 4.24) | 16.3 (± 3.54) | 15.9 (± 4.03) | 25.3 (± 5.28) |
| **Second Chamber** | | | | | |
| Width in cm | – | – | 3.2 (± 1.23) | 2.6 (± 0.83) | 4.1 (± 0.45) |
| Height in cm | – | – | 2.9 (± 0.89) | 2.3 (± 0.69) | 2.9 (± 0.65) |
| Volume in cm$^3$ | – | – | 21.1 (± 18.7) | 10.3 (± 9.69) | 25.4 (± 6.59) |
| Depth below ground in cm | – | – | 32.1 (± 7.03) | 30.0 (± 5.23) | 48.3 (± 10.04) |
| **Third Chamber** | | | | | |
| Width in cm | – | – | 3.3 (± 1.28) | 2.7 (± 0.57) | 3.4 (± 0.00) |
| Height in cm | – | – | 3.0 (± 1.18) | 2.3 (± 0.53) | 2.0 (± 0.00) |
| Volume in cm$^3$ | – | – | 23.6 (± 26.1) | 9.4 (± 6.67) | 12.1 (± 0.00) |
| Depth below ground in cm | – | – | 48.4 (± 9.71) | 44.8 (± 5.41) | 71.0 (± 8.27) |

In many nests, the first chamber was empty and did not contain any garden, whereas deeper chambers always had gardens (i.e., we did not find second and third chambers that were empty and without gardens; Excel Sheet 3, S1 Dataset). This suggests that deeper chambers in multi-chambered nests are used for fungiculture throughout the year, but the first chamber may be used for fungiculture only seasonally, for example during cooler and wetter seasons, but not during hot dry summers where the soil may have insufficient humidity at the depth of the first chamber. Brood is kept always embedded in gardens (i.e., never separate from gardens), which is typical for all mycelial-cultivating fungus-farming ants, and different from the yeast-cultivating *Cyphomyrmex* species that keep brood somewhat separate from their yeast gardens [16, 18, 65]. We found queens rarely in the first chamber even if garden was present there, and queens may therefore prefer the deeper chambers, which are typically larger, have larger gardens, and tend to have most of the brood in a given nest (Table 2; Excel Sheet 3, S1 Dataset).

Some nests also had a small chamber of about 1.5 cm diameter, at about 2–9 cm depth (average 5.6 cm depth), very close to the surface. We call this entrance chamber here 'vestibule' because this chamber never contained a full garden, with exception of one nest (AGH000620-01) that had a very minute garden hanging from the vestibule ceiling but this vestibule contained also some excavated sand. At the shallow depths of vestibules, gardens are expected to dry out and possibly overheat regularly during the predictably hot summers of central Texas, and such chambers therefore seem less suitable for fungiculture. Instead, the vestibule was

either empty, or contained recently excavated sand or expended garden refuse that workers temporarily stored in the vestibule before dumping the excavate or refuse outside the nest. We found excavated sand and refuse in the vestibule sometimes after rains, that is when the nest entrance had been temporarily washed shut, so workers were forced to moved and store excavated sand and garden refuse from deeper chambers into the vestibule, to dump excavate and refuse outside the nest as soon as the nest entrance could be reopened by the workers after a rain. The vestibule is not the remnant of the original foundress chamber (foundress chambers are deeper at ≈15 cm, whereas vestibules are at ≈5 cm depth; Table 2, Fig 4), but the vestibule is added later by workers as they expand the entrance tunnel a few centimeters below the surface. Many other fungus-farming ant species construct such vestibules in entrance tunnels for temporary storage of excavated soil, garden refuse, and sometimes gardening substrate such as dried leaves (UGM personal observation), or possibly as a staging area where workers can assemble to depart for foraging or defend a nest.

## Fungiculture

Fungus gardens of *M. hartmanni* are always suspended from the ceiling of the subterranean chambers. Gardens have the appearance of cream-colored, flocculent, mycelial curtains that are affixed to the sandy chamber ceiling or from rootlets (if present), without the garden curtains touching the side walls or floor of a chamber. The substrate used by *M. hartmanni* to nourish the cultivated fungus is a mix of fibrous bits of plant debris, flower parts such as anthers, seed husks, seeds, and arthropod feces, all knit together by mycelium of the cultivated fungus. One nest excavated in late September (UGM000930-02) had numerous alate wings embedded in the garden matrix together with the typical fibrous plant material and arthropod frass.

Under magnification, it is possible to see in the interior of *M. hartmanni* gardens sometimes lawns of gongylidia-like hyphal-tip swellings, as occurs also in many other lower-Attina species (UGM unpublished observations; [66]). These pyriform (pear-shaped) swellings in *M. hartmanni* gardens appear translucent and glassy, similar to the gongylidia of higher-Attina fungal cultivars [16, 20, 67, 68], but the glassy swellings in *M. hartmanni* gardens occur only in the interior of gardens, never at the outside garden surface facing the chamber walls (gongylidia of higher-Attina fungi grow also on the garden outside). In addition, the glassy swellings of *M. hartmanni* gardens do not occur in clusters of uniform round size as in the gongylidia clusters (staphylae) of higher-Attina fungal cultivars, but the glassy swellings are typically spread out as amorphous lawns, often in close proximity to brood. Wheeler (page 764 in [15]) likewise noticed in *M. hartmanni* gardens such concentrations of hyphal-tip swellings (called "bromatia" by Wheeler) and gives the measurements for "typical pyriform gongylida" as "1.5–4.3 μ in length and 1.3–4 μ in breadth". Wheeler's measurements appear incorrect by a factor of 10, as the diameter of hyphal-tip swellings in gardens of the lower-Attina *Mycocepurus smithi* range between 16–25 μm and in gardens of higher Attina between 20–60 μm [66].

Old expended garden is discarded by workers outside the nest, but *M. hartmanni* does not accumulate conspicuous refuse piles near the mounds as do many other species of fungus-growing ants, such as the sympatric *Cyphomyrmex wheeleri* (UGM, unpublished observations). In some nests, garden refuse is converted by workers into a dark wad of thick tar-like or clay-like consistency, and such wads can be found at the bottom of some (but not all) of the larger garden chambers underneath healthy, hanging gardens. The wads are packed tightly by workers and sculpted into smooth-surfaced, moist masses, so they are not loose piles of refuse strewn haphazardly across a chamber floor. Many other fungus-growing ant species from both the lower- and higher-Attina can have such tar-like wads ([39]; UGM, unpublished

observations), like the "wet mud pellet . . . at the bottom of the chamber, possibly consisting of dirt or refuse" found in some garden chambers of *Mycetophylax asper* [21]. Why Attina workers maintain these refuse wads inside garden chambers is not known.

Because the topmost garden chambers of *M. hartmanni* are rather shallow at 15–20 cm depth, and because *M. hartmanni* nests in open, sun-exposed habitat, the topmost fungus gardens must experience significant heat and reduced moisture during the peak summer months, but possibly also greater warmth during the day in winter and spring than deeper gardens do during those seasons. It is possible that *M. hartmanni* may therefore relocate gardens seasonally between different depths to optimize temperatures and moisture for garden growth and brood development, as occurs also in the sympatric leafcutter ant *Atta texana* [57], the higher-Attina *Trachymyrmex septentrionalis* in the south-eastern USA [69, 70], and in some subtropical leafcutter ants of South America [71, 72].

The fungus cultivated by *M. hartmanni* in central Texas is an undescribed species in the genus *Leucocoprinus* (Genbank accession AF079727) that is sequence-identical in the fast-evolving ITS gene to the fungus cultivated by the sympatric fungus-farmer *Cyphomyrmex wheeleri* (Genbank accessions JQ617602 & JQ617596; [26]) whose range overlaps with the range of *M. hartmanni* in south and central Texas, but not in east Texas (UGM unpublished observations; [3]). These two fungus-farming ant species of the lower Attina therefore share the same fungal cultivar type (likely the same fungus species; [20, 26], either because both ant species acquire fungal strains from the same free-living fungal populations occurring wild in North America, or because the two ant species exchange fungi on occasion by lateral transfer of garden, as is known for other fungus-growing ant species [19, 20, 73].

## Predation

We observed only one instance of predation on *M. hartmanni*. On 12. May 2000, we noticed that several of our flagged *M. hartmanni* nests, located in a more shaded area at the margin of our main study site at Stengl LPBS, had been overrun by an encroaching nest of the fire ant *Solenopsis invicta*. Such instances appear to be infrequent, however, perhaps because fire ants avoid sun-exposed dry areas [74], like the two forest clearings at our study site where *M. hartmanni* nests were most abundant. *Neivamyrmex* and *Labidus* army-ant species occur at Stengl LPBS and throughout the range of *M. hartmanni*, and these army ants are likely to prey on *M. hartmanni* as they do on other fungus-growing ant species [75–78], although we never observed army-ant predation on *M. hartmanni*.

## Conclusion

Our study is the first seasonal study of the rarely collected fungus-farming ant species *Mycetosoritis hartmanni*, and the first study to model colony longevity and quantify seasonal sex-ratio changes in any fungus-farming ant species. *M. hartmanni* is a sand specialist with nests of typically 40–60 workers and a simple nest architecture that enables easy excavation of nests. Because modeling indicates that nests that survive the first year have an average lifespan of an additional 1.9–2.5 years, and because 5% of these established colonies are expected to reach a lifespan of 5.7–7.4 years (Table 1), colony activity of single nests can be observed in a long-term study spanning many years, possibly for up to 8–10 years if nests are tracked from colony foundation. The protogynous production of reproductives during a three-month time window each year in summer, combined with the ease of excavating this fungus-farming ant species in sandy soil, recommends *M. hartmanni* for future sex-ratio studies in a fungus-farming ant species. *M. hartmanni* can be locally very abundant throughout its range, particularly in Texas. So far, invasive fire ants do not seem to have depressed abundances of *M. hartmanni* nests in the

sun-exposed sandy habitat preferred by this ant species, and at present *M. hartmanni* does not appear to be a threatened species in the USA. Future research could address how the nest architecture, survivorship, and sex-ratio patterns reported here for a population in Central Texas compare to other populations across the range of *M. hartmanni* from the USA to Central America.

## Supporting information

**S1 Dataset. Excel file summarizing raw data and metadata.**
(XLSX)

**S1 File. Colony survivorship modeling and R-script.**
(PDF)

## Acknowledgments

We are grateful to Phil Shappert, Steven Gibson, and Larry Gilbert for help and permission to conduct research at the Stengl Lost Pines Biological Station; and two anonymous reviewers for many helpful comments that improved the manuscript.

## Author Contributions

**Conceptualization:** Ulrich G. Mueller, Anna G. Himler.

**Data curation:** Ulrich G. Mueller, Anna G. Himler, Caroline E. Farrior.

**Formal analysis:** Ulrich G. Mueller, Anna G. Himler, Caroline E. Farrior.

**Funding acquisition:** Ulrich G. Mueller, Anna G. Himler.

**Investigation:** Ulrich G. Mueller, Anna G. Himler, Caroline E. Farrior.

**Methodology:** Ulrich G. Mueller, Anna G. Himler, Caroline E. Farrior.

**Project administration:** Ulrich G. Mueller.

**Resources:** Ulrich G. Mueller.

**Software:** Caroline E. Farrior.

**Supervision:** Ulrich G. Mueller.

**Validation:** Ulrich G. Mueller, Caroline E. Farrior.

**Visualization:** Ulrich G. Mueller, Caroline E. Farrior.

**Writing – original draft:** Ulrich G. Mueller, Anna G. Himler, Caroline E. Farrior.

**Writing – review & editing:** Ulrich G. Mueller, Anna G. Himler, Caroline E. Farrior.

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
