## [Decision Letter · Decision Letter 0]

20 Jan 2023

PONE-D-22-32328Life history, nest longevity, sex ratio, and nest architecture of the fungus-growing ant Mycetosoritis hartmanni (Formicidae: Attina)PLOS ONE

Dear Dr. Mueller,

Thank you for submitting your manuscript to PLOS ONE. After careful consideration, we feel that it has merit but does not fully meet PLOS ONE’s publication criteria as it currently stands. Therefore, we invite you to submit a revised version of the manuscript that addresses the points raised during the review process.

I agree with the reviewers that this is an interesting study that deserves publication, and I enjoyed the level of detail you provide. The reviewers suggested some edits that you might want to consider, and I’ll add my own:

The introduction seems a bit redundant and/or unorganised. For example, you explain Wheeler’s observations twice, once in the beginning and then later in more detail. The beginning of the introduction almost reads like an abstract, and some information in the introduction might fit better into the Methods section (e.g. references to the supplementary material). The explanation of the species name also feels a bit out of place since it follows a paragraph that made me want to see the methods and the data – perhaps the species name discussion can come earlier in the introduction or later in the discussion?

We look forward to receiving your revised manuscript.

Kind regards,

Volker Nehring

Academic Editor

PLOS ONE

Journal Requirements:

"The research was supported by funding from the National Science Foundation (Doctoral Dissertation Improvement Grant to AGH; CAREER award DEB-998379 and OPUS award DEB-1911443 to UGM); and the W.M. Wheeler Lost Pines Endowment from the University of Texas at Austin."

"No competing interests."

Reviewers' comments:

Reviewer's Responses to Questions

**Comments to the Author**

1. Is the manuscript technically sound, and do the data support the conclusions?

Reviewer #1: Yes

Reviewer #2: Yes

2. Has the statistical analysis been performed appropriately and rigorously? 

Reviewer #1: I Don't Know

Reviewer #2: Yes

3. Have the authors made all data underlying the findings in their manuscript fully available?

Reviewer #1: Yes

Reviewer #2: Yes

4. Is the manuscript presented in an intelligible fashion and written in standard English?

Reviewer #1: Yes

Reviewer #2: Yes

5. Review Comments to the Author

Reviewer #1: GENERAL OPINION:

The manuscript gives a nice (but in some cases incomplete, see below) overview about the biology of the poorly known fungus-farming Mycetosoritis hartmanni ant, mostly about its life history, longevity, nest architecture, phenology, sex ratio and survivorship. I’m basically happy to read such a detailed “naturalist” description about a species’ biology and hope that this work will find its place in a high quality journal for publishing. Despite such knowledge on different species would be very important in the days of the climatic catastrophe, the “modern” stories are rather fashionables nowadays. I it should be the duty of the editors to consider whether such a story could be enough interesting for the PLOS ONE’s readers. It will be, as I hope. In the case of refusing, I would encourage the authors to submit a rewritten version of the manuscript to some journal read basically by myrmecologists.

The main research has been done about twenty years ago and I feel somewhere that the general description is not up-to-date. To tell the truth, despite I’m a myrmecologist who does not work with Attina ants, I clearly remember for some phylogenetic trees of this group. So, why the phylogenetic position of this ant is not discussed shortly in such a work which could be a review on the knowledges of a poorly known species? On the other hand, there are some points that are mentioned in several places, see some examples below. It would be nice to read critically the full text by the authors and live the repeating thoughts only on the most relevant place.

MAJOR ISSUES:

-Line 1, §1, Page 3: could the authors give the type species of the genus?

-§2, Page 3 and/or some other part(s): I miss a more general overview about the known biology of this ant, in some short sentences. E.g. what about its colony founding, dietary, preferred climate and vegetation, and maybe about the topics discussed in this manuscript in detail? I know that such details are discussed in the manuscript sporadically but it would be better to see in the Introduction the summarised knowledge on this ant already known before this work.

-Last §, P 3: It’s nice that the nomenclature of this species is discussed. In this case the taxonomic name (“Mycetosoritis hartmanni (Wheeler, 1907)”) and the original name (“Atta hartmanni Wheeler, 1907”) also should be mentioned. Officially, the taxonomic name should be given in the 1st mentioning of the species in a work (currently at L6, P2).

-Last sentence, §1, P4: “ICZN 1999. International Code of Zoological Nomenclature. Fourth Edition. The International Trust for Zoological Nomenclature, London, UK. 306 pp.” (https://www.vliz.be/imisdocs/publications/271138.pdf) should be checked and cited.

-1st sentence of Materials and Methods: The authors basically worked on a single, about northernmost population of a relatively widely spread (from Louisiana-Texas to Honduras) species and later did some general conclusions on the biology of this ant. It would be important to emphasize in the Conclusion that the biology of other populations could be partly different, e.g. at the southernmost regions. See e.g. L10-13, §3, P14 for a population with different biology.

-About active or passive: The authors use active formula too much, as I feel (e.g. Last lines, P5: The authors start four sentences with “We” in a methodological part): I’m not a native English speaker and not sure how PLOS ONE prefers the frequent usage of active formula. However, I think the passive structure is better at drawing attention to the research, while the active structure is better at expressing the authors' own opinions.

-About citing: When there is a statement, its background should be clearly given which can be some citation and/or the results of the current work. E.g. the authors write withouth citation that “…mortality rates of small and incipient nests are typically high in ant species.” (L9, §2, P7-8). Is it a thought? Or there are some minor cases, like (10-11, §2, P12): “Because July and August are the hottest and driest months in central Texas, with more regular rains resuming typically in late August or early September…”

-About the survival of the colonies (e.g. L last before, §2, P7): are the authors sure in that all the colonies have died of and not moved to some other places (e.g. because the changing of the vegetation, arriving of some aggressive ant colonies, etc.)? Please, discuss it.

-About “queen may run out of stored sperm” (e.g. L7, §2, P8; L1, §1, P12 and in other parts): I feel it to be a bad speculation, especially in the case of a species which has so small colonies. Can the authors imagine that only few hundred-few thousands of sperms a queen receives during copulation? Do the authors know some similar stories at some other ant species? Could they cite them?

-§1 of Results and Discussion: Could the authors give the total number of nests they examined?

-It would be nice to see some photos about the ant and the nest in such a descriptive works. The authors cite some from iNaturalist (L5, §2, P9) but its stability is not guaranteed.

-About repetitions: e.g. it was not a new information for me, after reading the manuscript up to this part (PP15-18, §2, P9): “The steep entrance tumuli identifying M. hartmanni nests are easiest to find after a time of no rain during the second half of spring when colonies excavate most sand during nest expansion. This explains why we found during general ant collecting throughout Texas additional M. hartmanni populations frequently in spring, and less during other seasons.” I had the similar feeling when I read (L11-12, §2, P10): “The largest nest had 148 workers, but worker number per nest is typically less than 100 workers in M. hartmanni.” So, could the authors be more critical with repetition thoughts?

-End of §1, P10: Do we know something about the history and biogeography of the two discussed genera? So, maybe Mycetosoritis arrived from a tropical region and Mycetophylax from a continental one which may explain the difference about the wintering habit of these ants.

-About careful drafting: e.g. it is written (L1-2, §2, P10) that “…most colonies had between 20-70 workers, only two colonies had over 100 workers (118 and 148 workers)…”. It would be right e.g. as: “…most colonies had between 20-70 COUNTED workers, only two colonies had over 100 COUNTED workers (118 and 148 workers)…”. What about the foraging workers and the ones which escaped, etc.? Be more careful with such strong statements, please. Similarly, “…with nests of USUALLY 40-60 workers…” (L3, §1, P18) would be the right form because the authors mention e.g. 148 workers too.

-L6, §2, P10: Did these incipient colonies have brood?

-About “non-claustral nest founding” (e.g. P11): “SEMI-claustral COLONY founding” is better, as I know.

-L1-2, §2, P11: Why the lifespan of that queen has not been monitored during the years to really see the lifespan at least one queen?

-L4-6, §2, P12: Could the authors summarise the total developing time?

-L16-17, §2, P12: “…suggesting that the foundress queen of the latter incipient nest had lost her garden…” Or has he not yet raised it?

-End of §1, P13: “This annual transition in spring from investment into colony growth and maintenance (investment into workers) versus into reproduction could be under queen control, but is more likely under worker control because workers would seem to have greater influence over the rearing conditions (e.g., nutrition, temperature) of developing female larvae.” Could it be compared with some other ant species, or discussed in detail in some other way?

-L10-11, §2, P15: “…queens may therefore prefer the deeper chambers…” And/or have they retreated there at the disturbing?

-P16-17: Could the fungus be given on some taxonomic level? The Genbank accessions are good but it would be nice to know something about the taxonomy of the fungus during reading.

MINOR ISSUES:

-Line numberings would improve the discussion about the manuscript.

-Abstract: “(average 47.6 ± 31.9 SD, maximum 148 workers)” can be deleted, such details are not crucial for an abstract.

-L7, §2, P2 and in several other cases: The sentence starts with abbreviated genus name. I’m not sure whether it’s a preferred format by the journal, many other journal do not like this style.

-L11, §2, P2: dots are missing after “M” in two cases

-L2 vs. L4, §2, P7: Maybe I do not understand it well but how 150 nests can be “additional” which already have been “pre-flagged”?

-L1-2, §1, P8: Repetitions: “…differences generated small differences in our mortality estimates. Estimates…”

Reviewer #2: The authors present a straightforward and engaging natural history of the seldom encountered fungus-farming ant, Mycetosoritis hartmanni. This study represents the only quantitative account of seasonal behavior, nest architecture, reproduction, and colony longevity in this species. I appreciated the detailed explanation of the taxonomic history of this species and the account of Wheeler’s early encounters with it. Especially interesting was the authors’ observation of changes in sexual investment across the breeding season and the estimate of colony lifespan. The conclusions are well-reasoned and follow the data well. A few minor changes and additions would improve this manuscript and increase its utility for other biologists.

My main critique: You repeatedly mention that colonies selected for your study could be operationally defined as “mature” because they were large and appeared to be more than one year old. For most ant species, there is a minimum number of workers (threshold) necessary for reproduction (which defines sexual maturity). Because you excavated and censused colonies with sexual present in some seasons, I think you likely have information about the minimum number of workers necessary for colony reproduction/maturity – but you have not reported this. It is my strong preference that the term ‘mature’ only be used if sexual were observed or if the minimum worker number (proxy) was observed outside of the mating season. If you don’t want to define the colonies by their actual size/worker number, I would call them something else: perhaps “large”, “active” or “visible.” We know from studies of at least 3 Pogonomymex species that some colonies maystay small for many years, and never reproduce (probably because of resource limitation or genetic incompatibilities with mates). Size, age, and maturity should not be conflated.

Materials and Methods:

1. “The focal population consisted of several hundred M. hartmanni nests distributed across two adjacent clearings in semi-open pine-oak forest.”

Please list the dominant pine and oak species present, as well as the more abundant, co-occurring ant species. Doing so will greatly increase the utility of this manuscript for future biologists seeking populations of M. hartmanni.

2. “because wet weights are better estimates of resource investment by ant colonies into reproductives than their dry weights.”

I disagree with the above statement. Many studies have found important seasonal and age-specific investment strategies by analyzing the proportion of lipids (extracted with organic solvents) + lean weight (dry, fat free), which provide more nuance than wet weight alone - especially among ants that divide labor between interior and exterior roles in arid environments. You might rephrase to say “ Wet weight is one accepted way of measuring resource investment by ant colonies and can act as a proxy for….”.

3. “Any queen senescence and sperm shortage in M. hartmanni unlikely affects colony survivorship already at our estimated average….”

Typo: change to “……is unlikely to affect colony survivorship…”

Fire ants are especially efficient with sperm use (just 3 sperm cells per fertilization event). However, queens do run out of sperm around 6 years of age – suggesting that sperm limitation may affect colony longevity for other ant species as well (see work of Tschinkel). We don’t know how much sperm your focal species stores, because you did not dissect the spermatheca. Therefore, I am not sure your statement is justified.

Results/Discussion:

4. Did you excavate colonies before foragers departed each day? If not, you might mention that colony sizes are estimates that may be conservative because of absent foragers.

5. Regarding colony reproduction: Is there a single, coordinated mating flight or multiple mating flights per season? If there were multiple mating flights, were sexuals collected between events or before the first flight? I may have missed your description of the mating flights.

6. What was the ratio of sexuals to workers in your colonies (Mean and standard deviation)? Did larger colonies produce more sexuals? Did colony size influence sex ratio? You have lots of untapped data here.

7. “The size of the third chamber likely depended on the age of a nest, with younger nests having smaller third chambers because the workers added a third chamber only recently to a nest and this third chamber had yet to be enlarged by the workers.”

Are you certain that all of these colonies were younger, and not just smaller?

8. Your description of the garden was a joy to read. It is beautifully described. I was left longing for a photograph. Consider adding an image of an excavated garden and the ants.

9. “It is possible that M. hartmanni may therefore relocate gardens seasonally between different depths to optimize temperatures and moisture for garden growth and brood development, as occurs also in the sympatric leafcutter ant Atta texana (Mueller et al. 2011) and some subtropical leafcutter ants of South America (Lapointe et al. 1998; Bollazzi et al. 2008).”

Jon Seal also showed annual, vertical movement of fungus gardens in Trachymyrmex septentrionalis. (Seal and Tschinkel 2006)

10. “On 12. May 2000, we noticed that several of our flagged M. hartmanni nests, located in a more shaded area at the margin of our main study site at Stengl LPBS, had been overrun by an encroaching nest of the fire ant Solenopsis invicta. Such instances appear to be infrequent, however, perhaps because fire ants avoid sun-exposed dry areas, like the two

forest clearings at our study site where M. hartmanni nests were most abundant”

- Another relevant citation here may be Josh King’s PNAS paper about how fire ants do not displace native ants and do not survive/invade in undisturbed habitat.

11. Fig 2: “Average sex ratios were calculated by first calculating the sex ratio for each

individual nest, then averaging sex ratios across all the nests collected within a 2-week time period (data in columns O & P in Excel Sheet 1 in Supporting Information).”

-Consider reporting variance for the averages.

6. PLOS authors have the option to publish the peer review history of their article (what does this mean?). If published, this will include your full peer review and any attached files.

Reviewer #1: No

Reviewer #2: No

---

## [Author Response · Author response to Decision Letter 0]

26 Apr 2023

We address all editor & reviewer comment in the letter Response to Reviewers submitted with this revised manuscript.

---

## [Decision Letter · Decision Letter 1]

31 May 2023

PONE-D-22-32328R1Life history, nest longevity, sex ratio, and nest architecture of the fungus-growing ant Mycetosoritis hartmanni (Formicidae: Attina)PLOS ONE

Dear Dr. Mueller,

Thank you for submitting your manuscript to PLOS ONE. After careful consideration, we feel that it has merit but does not fully meet PLOS ONE’s publication criteria as it currently stands. Therefore, we invite you to submit a revised version of the manuscript that addresses the points raised during the review process.

I enjoyed the manuscript and ruled this a minor revision to give you the opportunity to consider the last open comments - this might be difficult in the proof stage.

We look forward to receiving your revised manuscript.

Kind regards,

Volker Nehring

Academic Editor

PLOS ONE

Journal Requirements:

Reviewers' comments:

Reviewer's Responses to Questions

**Comments to the Author**

1. If the authors have adequately addressed your comments raised in a previous round of review and you feel that this manuscript is now acceptable for publication, you may indicate that here to bypass the “Comments to the Author” section, enter your conflict of interest statement in the “Confidential to Editor” section, and submit your "Accept" recommendation.

Reviewer #1: All comments have been addressed

Reviewer #2: All comments have been addressed

2. Is the manuscript technically sound, and do the data support the conclusions?

Reviewer #1: Yes

Reviewer #2: (No Response)

3. Has the statistical analysis been performed appropriately and rigorously? 

Reviewer #1: I Don't Know

Reviewer #2: (No Response)

4. Have the authors made all data underlying the findings in their manuscript fully available?

Reviewer #1: Yes

Reviewer #2: (No Response)

5. Is the manuscript presented in an intelligible fashion and written in standard English?

Reviewer #1: Yes

Reviewer #2: (No Response)

6. Review Comments to the Author

Reviewer #1: Thank you very much for considering my suggestions. I accept all of your answers and have only two minor requests. After managing these, I think this MS does not need some further review procedure:

1) Lines 64-65: The taxonomic names are not in a right form. The describer and year should be separated by a coma. Furthermore, when a species was described in another genus than the recently valid one, the describer and year should be closed to parentheses. So, the valid names are (as I gave in my comment): “M. hartmanni (Wheeler, 1907)” and “M. vinsoni Mackay, 1998”. See, e.g., antweb.org.

2) Lines 306-312 vs. 532-543: there are many repetitions. I think, it would be enough to write down these speculative thoughts only once and write a “see below/above” or “see Colony survivorship/Absence or rarity of colony migration” at the deleted part.

Reviewer #2: (No Response)

7. PLOS authors have the option to publish the peer review history of their article (what does this mean?). If published, this will include your full peer review and any attached files.

Reviewer #1: No

Reviewer #2: No

---

## [Author Response · Author response to Decision Letter 1]

6 Jul 2023

EDITOR COMMENTS

I enjoyed the manuscript and ruled this a minor revision to give you the opportunity to consider the last open comments - this might be difficult in the proof stage.

REVIEWER COMMENTS

1. If the authors have adequately addressed your comments raised in a previous round of review and you feel that this manuscript is now acceptable for publication

Reviewer #1: All comments have been addressed

Reviewer #2: All comments have been addressed

6. Review Comments to the Author

Reviewer #1: Thank you very much for considering my suggestions. I accept all of your answers and have only two minor requests. After managing these, I think this MS does not need some further review procedure:

1) Lines 64-65: The taxonomic names are not in a right form. The describer and year should be separated by a coma. Furthermore, when a species was described in another genus than the recently valid one, the describer and year should be closed to parentheses. So, the valid names are (as I gave in my comment): “M. hartmanni (Wheeler, 1907)” and “M. vinsoni Mackay, 1998”. See, e.g., antweb.org.

2) Lines 306-312 vs. 532-543: there are many repetitions. I think, it would be enough to write down these speculative thoughts only once and write a “see below/above” or “see Colony survivorship/Absence or rarity of colony migration” at the deleted part.

Reviewer #2: (No Response)

OUR RESPONSE: We have addressed the two comments by Reviewer 1:

(a) We now use in the third paragraph of the Introduction the taxonomically correct forms of the species “M. hartmanni (Wheeler, 1907)” and “M. vinsoni Mackay, 1998”, per recommendation by Reviewer 1.

(b) We edited lines 306-312 vs. 532-543 to reduce the appearance of repetitions. We added a cross-reference “as already noted by Wheeler (1907) (see above)” to the section in lines 306-312; and we deleted from lines 532-543 the subclause “that most nests accumulate in spring when workers expand nests after inactivity in winter” because that same information is already presented in the Introduction. We are hesitant to delete additional information, and we feel the writing in these two sections is appropriate and helpful to the reader.

We correct in lines 342 and 388 two minor typos that we overlooked in our previous revision (these corrections are highlighted in the PDF recording in TrackChanges all our edits since the last revision).

We also double-checked all the references; each citation appearing in the main text is listed among the References, and there is no redundant publication listed in the References that is not cited in the main text.

---

## [Editor Report · Decision Letter 2]

13 Jul 2023

Life history, nest longevity, sex ratio, and nest architecture of the fungus-growing ant Mycetosoritis hartmanni (Formicidae: Attina)

PONE-D-22-32328R2

Dear Dr. Mueller,

We’re pleased to inform you that your manuscript has been judged scientifically suitable for publication and will be formally accepted for publication once it meets all outstanding technical requirements.

Kind regards,

Volker Nehring

Academic Editor

PLOS ONE

Additional Editor Comments (optional):

Thanks for submitting this, I really like the study and the attention to detail!
---

## [Editor Report · Acceptance letter]

18 Jul 2023

PONE-D-22-32328R2 

Life history, nest longevity, sex ratio, and nest architecture of the fungus-growing ant *Mycetosoritis hartmanni* (Formicidae: Attina) 

Dear Dr. Mueller:

I'm pleased to inform you that your manuscript has been deemed suitable for publication in PLOS ONE. Congratulations! Your manuscript is now with our production department. 

Kind regards, 

on behalf of

Dr. Volker Nehring 

Academic Editor

PLOS ONE